# Novel NFκB Inhibitor SC75741 Mitigates Chondrocyte Degradation and Prevents Activated Fibroblast Transformation by Modulating miR-21/GDF-5/SOX5 Signaling

**DOI:** 10.3390/ijms222011082

**Published:** 2021-10-14

**Authors:** Pei-Wei Weng, Vijesh Kumar Yadav, Narpati Wesa Pikatan, Iat-Hang Fong, I-Hsin Lin, Chi-Tai Yeh, Wei-Hwa Lee

**Affiliations:** 1Department of Orthopaedics, School of Medicine, College of Medicine, Taipei Medical University, Taipei City 110, Taiwan; wengpw@tmu.edu.tw; 2Department of Orthopaedics, Shuang Ho Hospital, Taipei Medical University, New Taipei City 23561, Taiwan; 3Graduate Institute of Biomedical Materials and Tissue Engineering, College of Biomedical Engineering, Taipei Medical University, Taipei City 110, Taiwan; 4Department of Medical Research & Education, Taipei Medical University-Shuang Ho Hospital, New Taipei City 23561, Taiwan; vijeshp2@gmail.com (V.K.Y.); narpatisesa@gmail.com (N.W.P.); 18149@s.tmu.edu.tw (I.-H.F.); ctyeh@s.tmu.edu.tw (C.-T.Y.); 5School of Dentistry, College of Oral Medicine, Taipei Medical University, Taipei City 110, Taiwan; 6Division of Periodontics, Department of Dentistry, Taipei Medical University-Shuang Ho Hospital, New Taipei City 23561, Taiwan; 7Department of Medical Laboratory Science and Biotechnology, Yuanpei University of Medical Technology, Hsinchu City 30015, Taiwan; 8Department of Pathology, Taipei Medical University-Shuang Ho Hospital, New Taipei City 23561, Taiwan

**Keywords:** osteoarthritis (OA), NFκB inflammatory circuit, miR-21/GDF-5/SOX5 signaling, NFκB inhibitor, therapeutics

## Abstract

Osteoarthritis (OA) is a common articular disease manifested by the destruction of cartilage and compromised chondrogenesis in the aging population, with chronic inflammation of synovium, which drives OA progression. Importantly, the activated synovial fibroblast (AF) within the synovium facilitates OA through modulating key molecules, including regulatory microRNAs (miR’s). To understand OA associated pathways, in vitro co-culture system, and in vivo papain-induced OA model were applied for this study. The expression of key inflammatory markers both in tissue and blood plasma were examined by qRT-PCR, western blot, immunohistochemistry, enzyme-linked immunosorbent assay (ELISA) and immunofluorescence assays. Herein, our result demonstrated, AF-activated human chondrocytes (AC) exhibit elevated NFκB, TNF-α, IL-6, and miR-21 expression as compared to healthy chondrocytes (HC). Importantly, AC induced the apoptosis of HC and inhibited the expression of chondrogenesis inducers, SOX5, TGF-β1, and GDF-5. NFκB is a key inflammatory transcription factor elevated in OA. Therefore, SC75741 (an NFκB inhibitor) therapeutic effect was explored. SC75741 inhibits inflammatory profile, protects AC-educated HC from apoptosis, and inhibits miR-21 expression, which results in the induced expression of GDF-5, SOX5, TGF-β1, BMPR2, and COL4A1. Moreover, ectopic miR-21 expression in fibroblast-like activated chondrocytes promoted osteoblast-mediated differentiation of osteoclasts in RW264.7 cells. Interestingly, in vivo study demonstrated SC75741 protective role, in controlling the destruction of the articular joint, through NFκB, TNF-α, IL-6, and miR-21 inhibition, and inducing GDF-5, SOX5, TGF-β1, BMPR2, and COL4A1 expression. Our study demonstrated the role of NFκB/miR-21 axis in OA progression, and SC75741’s therapeutic potential as a small-molecule inhibitor of miR-21/NFκB-driven OA progression.

## 1. Introduction

Osteoarthritis (OA) is a complex joint disease that mostly occurs in the aging population [1], and it is characterized by cartilage degeneration, osteophyte synthesis, and subchondral sclerosis, as well as synovial inflammation and thickening [2,3,4,5]. The pathogenesis of OA involves “wear and tear” and tissue degradation followed by tissue repair in response to transforming growth factor-beta (TGF-β)–induced myofibroblast production in the extracellular matrix (ECM) [6]. Chondrogenesis is one of the earliest phases of skeletal development [7] and compromised chondrogenesis causes OA development and progression. Bone morphogenetic proteins (BMPs) and growth differentiation factor 5 (GDF-5) also play critical roles in chondrocyte biology [8,9]. Along with BMPs and enzymes in the ECM, collagens participate in the regulation of cartilage remodelling and OA progression [10]. In early-stage OA, synovial inflammation develops into synovitis [11], which directly contributes to several clinical signs and symptoms, including swelling and structural changes in the joints [2,11,12]. Synovitis-associated pathological alteration of the joints causes the release of proinflammatory cytokine mediators, such as interleukin 1β (IL-1β) and tumor necrosis factor-α (TNF-α) [13]. These inflammatory mediators are produced by several types of cells in the joints, including synovial fibroblasts (SFs) [14,15], the dominant cells in the synovium [16]. Activated synovial fibroblasts (AFs) in the synovium regulate the influx and efflux of leukocytes and are involved in cartilage alteration [17], which in turn creates a vicious cycle that amplifies synovial inflammation through the induction of certain signaling molecules, damaging the cartilage and eventually the bone [11]. Studies have indicated that mechanical stress and elevated levels of proinflammatory cytokines in OA-affected joints disrupt osteoclast differentiation [18] and play a causative role in upsetting the critical balance between anabolic and catabolic signals in cartilage homeostasis [19,20]. Despite the increase in the prevalence of OA, its pathogenesis is not studied well.

Interestingly, the patients with symptomatic OA show pain, while asymptomatic OA patients do not display pain, but show radiographic signs of joint damage; the role of the infrapatellar pad (IPFP), other than the synovial membrane, has been reported as a potential reason for pain in OA [21]. The adipocytes, preadipocytes, macrophages, fibroblasts, and other associated cells in the IFP secrete adipokines, cytokines, and growth factors that can influence the internal homeostatic balance of joints [22]. Studies have indicated the IPFP and synovial membrane might act as a functional unit in the pathogenesis of OA and pain [21].

Abnormally activated nuclear factor-κβ (NFκB) is a key player in OA progression through chondrocyte catabolism and synovial inflammation [23,24,25]. Thus, regulators or cofactors of NFκB that can serve as potential targets in OA interventions should be identified [23,26]. Studies have revealed that microRNAs (miR’s), which are conserved 18–25-nucleotide endogenous-noncoding RNAs, are molecules with enzymatic and regulatory roles in synovitis and synovial hyperplasia [27,28]. A recent study of articular cartilage explants stimulated by IL-1β observed that miR’s were differentially expressed in the knee synovial fluid of patients with OA [29]. Many studies have reported that altered levels of miR’s, including miR-885-5p, miR-195, and miR-126 occur in the plasma of patients with OA [13,30]. The reduced expression of miR-9 and miR-98 is indicative of their antiapoptotic activity and their enhancement of chondrocyte proliferation through NFκB expression regulation [25,29]. NFκB proteins constitute a family of inducible transcription factors which modulate many key genes involved in the pathogenesis of inflammatory diseases and are highly expressed in patients with OA. This study explored the therapeutic effect and efficacy of SC75741 [31], a potent inhibitor of NFκB, on OA and the suppression of the expression of inflammatory cytokines, chemokines, and proapoptotic factors as the underlying mechanism [32].

To the best of our knowledge, this is the first prospective study suggesting that AF-educated healthy chondrocytes (HCs) exhibit elevated levels of the inflammatory markers NFκB, TNF-α, IL-6, and miR-21. Coculturing with AFs resulted in the increased apoptosis of HCs and significantly reduced the expression of chondrogenesis inducers, namely SOX5, TGF-β1, and GDF-5. SC75741 treatment significantly suppressed miR-21 levels and increased the expression of its targets GDF-5, SOX5, TGF-β1, BMPR2, and COL4A1. Our in vivo study of OA demonstrated that SC75741 treatment significantly reduced the destruction of the articular joint; suppressed the levels of the inflammatory markers NFκB, TNF-α, IL-6, and miR-21; and increased the levels of the chondrogenic inducers GDF-5, SOX5, TGF-β1, BMPR2, and COL4A1. These findings provide insights into the role of the NFκB inflammatory circuit and miR-21 in OA progression as well as the therapeutic potential of SC75741 as a potent chondroprotective agent and small-molecule inhibitor of miR-21-/NFκB-driven OA progression.

## 2. Results

### 2.1. IL-1β-Activated Synovial Fibroblasts Exhibit Amplified Inflammatory Signaling in Chondrocytes

We first simulated the arthritic synovial microenvironment by culturing a fibroblast-like SW982 synovial cell line with 10 ng/mL of pro-inflammatory cytokines, IL-1β for 24 h. The resultant SW982 cells (hereafter referred to as AF) showed markedly increased expression of α-SMA, a marker for activated fibroblast, compared with their unstimulated control counterparts (Figure 1A). Subsequently, human chondrocyte cell lines, C28/I2 and CA402, were cultured in the presence of AF, referred to as activated chondrocytes (AC). AF co-culture chondrocytes, i.e., in AC cells demonstrated significantly increased expression of inflammatory markers IL-6, NFκB, TNF-α, and Cox2 at the mRNA and protein levels (Figure 1B,C), suggesting a role for AF in promoting chronic inflammation. We then searched publicly available GEO (GSE37627) databases for candidate miR’s associated with dysfunctional fibroblasts, including activated fibroblasts. As indicated, miR-21, miR-422a, and miR-619 were found to be the top three miRs identified in dysfunctional fibroblasts (Figure 1D). Using an online miRTargetLink Human target search tool [33], we found that miR-21 targets genes that play key roles in different aspects of chondrogenesis, namely SOX5, GDF-5, TGFBR2, COL4A1, and SMAD7 (Figure 1E). Consistently, we found an elevated level of miR-21 in both C28/I2 and CA402 cells after AF induction (Figure 1F).

### 2.2. miR-21 Modulates the Activity of Activated Fibroblast-like Chondrocytes

Next, we tested our hypothesis that miR-21 modulates the transformation of activated fibroblast-like SW982 (AFs) cells. We demonstrated that while incubation with IL-1β or miR-21 mimics promoted the AF trait of SW982 cells as demonstrated by increased α-SMA protein expression, treatment with the miR-21 inhibitor alone or in the presence of IL-1β markedly suppressed the expression of α-SMA protein (Figure 2A,B). Besides, SW982 cells bearing miR-21 mimic showed increased migratory ability while migration was significantly suppressed in their counterparts with the miR-21 inhibitor (Figure 2C). Furthermore, co-culturing with miR-21 inhibitor-treated AF-induced C28/I2 and C20A4 cells exhibited induced expression levels of chondrogenesis markers, namely GDF-5, SOX5, TGF-β1, COL4A1, and ACAN proteins in comparison with the miR-21 mimic and NC control cells (Figure 2D).

### 2.3. Ectopic Expression of miR-21 in Fibroblast-like Activated Chondrocytes Promote Osteoblast-Mediated Differentiation of Osteoclasts In Vitro

In the progressive development of OA, the balance between bone formation and resorption is significantly skewed towards the latter [34]. Our investigation of the effects of miR-21 overexpression in AFs on osteoblast differentiation showed that co-culture of RAW264.7 cells with the activated fibroblast-like SW982 chondrocytes pre-exposed to miR-21 mimic significantly increased the population of macrophage-lineage TRAP+ cells compared with their control or miR-21 inhibitor-exposed counterparts (Figure 3A), suggesting that ectopic expression of miR-21 promotes the osteoblast-mediated differentiation of osteoclast precursors to osteoclasts. The methods suggested by Lampiasi, Nadia et. al, 2021 [35] were applied in our study for osteoclasts differentiation from murine RAW264.7 cells stimulated by RANKL. Western blot analyses showed that compared with negative control (NC) or RAW264.7 cells co-cultured with miR-21 inhibitor-treated AFs, RAW264.7 cells co-cultured with miR-21 mimic pre-treated AFs showed upregulated expression of osteoclast differentiation markers, matrix metallopeptidase (MMP) 9, receptor activator of NFκB ligand (RANKL), triiodothyronine receptor auxiliary protein (TRAP), and osteoprotegerin (OPG) proteins (Figure 3B). Moreover, we demonstrated that co-culture with miR-21 mimic pre-treated AFs significantly enhanced the proliferation of RAW264.7 cells compared with their counterparts in the control group or co-cultured with miR-21 inhibitor-treated ACs (Figure 3C). Probing the cell-type independence of miR-21 effect, we observed that exposure of RAW264.7 cells to miR-21 mimic markedly increased the number of TRAP+ cells compared with the control or miR-21 inhibitor-treated group (Figure 3D), suggesting that the ectopic expression of miR-21 in RAW264.7 cells also enhances the tendency towards osteoclast differentiation.

### 2.4. Inhibition of NF-κβ Dampens Cellular Transformation of Activated Fibroblasts, Chondrocytes, and Osteoclasts

NFκB is a principal factor in chronic inflammation and progression of osteoarthritis [36,37]. In addition, miR-21 has been shown to induce NFκB activity and promote NFκB signaling by suppressing PTEN and activating Akt [38,39]. Thus, we explored the therapeutic potential of the NFκB inhibitor SC75741 in the cellular model of OA. Exposure of AFs with 2 µM SC75741 for 48 h resulted in markedly reduced co-expression of α-SMA and NFκB proteins (Figure 4A, left). Concomitantly, SC75741-treated AFs exhibited a significantly reduced expression of NFκB, α-SMA, TNFα, and IL-6 at protein level (Figure 4A, right up) and reduced expression of miR-21 (3.41-fold, *p* < 0.001) (Figure 4A, right down) was also observed. Next, we observed that co-culturing SC75741-treated ACs cells (C28/I2 or C20A4 chondrocytes) elicited significant downregulation of protein markers of hypertrophic chondrocytes, namely GDF-5, SOX5, COL4A1, ACAN, and TGF-β1 (Figure 4B, left) as well as suppressed the expression of miR-21 (Figure 4B, right). Furthermore, when treated with 2 µM SC75741 overnight, the population of TRAP+ AFs RAW264.7 cells was significantly reduced (5.3-fold, *p* < 0.001) (Figure 4C), and this were associated with downregulated expression of MMP9, RANKL, TRAP, and OPG proteins (Figure 4D), suggesting markedly reduced capability to induce osteoblast differentiation.

### 2.5. In Vivo Evaluation of the Therapeutic Potential of SC75741 in the OA Rat Model

After demonstrating that the treatment with SC75741 resulted in reduced generation of activated chondrocytes, and osteoclasts in vitro, we evaluated its potential therapeutic effects in vivo using the rat OA model. The rat OA model was established by intra-articular injection of the proteolytic enzyme, papain, and the histopathological changes in the articular structure were assessed by hematoxylin and eosin (H&E) staining and Osteoarthritis Research Society International (OARSI) scores. In comparison with the control rats with normal joints, we observed marked destructive changes in the articular cartilage morphology of the papain-induced OA models, including cartilage erosion, proteoglycan accumulation, and cellular loss or denudation; this OA-associated articular destruction was markedly ameliorated by treatment with SC75741 (15 mg/kg) (Figure 5A). Importantly, the SC75741-induced reduction in the severity of OA cartilage destruction correlated with a significantly lower OARSI score (Figure 5B). This is consistent with immunohistochemical staining results that showed that SC75741 (15 mg/kg) treatment significantly downregulated the expression of active Caspase-3 protein, a marker of apoptosis in the rat articular cartilage compared with the untreated OA model or their normal counterparts (Figure 5C). Interestingly, as demonstrated by Q-score values, we observed that treatment with SC75741 also markedly reduced the protein expression of MMP9, IL-6, NFκB, TNF-α, and IL-1β (Figure 5D), as well as the plasma and tissue miR-21 expression level (Figure 5E) in comparison with the untreated OA group.

### 2.6. In Vivo SC75741 Inhibited Chondroclast Differentiation and Activation

Treatment with SC75741 significantly downregulated the expression of TRAP and RANKL in the joints and cartilage of OA rats (Figure 6A). To assess the effect of SC75741 on RANKL induced chondroclast differentiation, the co-culture system was applied as suggested previously by Kwon, J.Y. et al. 2018 [40]. The bone marrow-derived monocytes (BMMs) were treated with SC75741 in the presence of M-CSF and RANKL for two weeks; we demonstrated that treatment with SC75741 significantly suppressed the formation of TRAP+ mature multinuclear chondroclasts (Figure 6B). SC75741-treatment results in the inhibition of NFκB circuit and suppressed miR-21 signaling in the activated chondrocytes, which conversely results in the upregulation of miR-21 predictive targets SOX5, TGF-β1, GDF-5, and COL4A1, (Figure 6C). Overall these results suggest that SC75741 has a significant inhibitory effect on RANKL-mediated chondroclast differentiation and activation.

## 3. Discussion

Inflammation plays a critical role in the pathogenesis of OA [2,3,4,5], a common disease that involves “wear and tear” and inflammation of the joints [6]. Our results indicate that AFs contribute to chronic inflammation through the activation of the NFκB inflammatory circuit and miR-21/GDF-5/SOX5 signaling within the synovium, which consequently promotes OA progression. The present study provides both in vitro and in vivo evidence that the administration of SC75741 ameliorates OA severity and inhibits its progression. Specifically, SC75741 treatment suppresses the expression of inflammatory markers and inhibits the apoptosis of AF-educated HCs. Conversely, it also induces chondrogenesis and increases the expression of COL4A1 and aggrecan (ACAN). OA causes joint inflammation and varying extents of joint dysfunction, as well as the loss of function and even disability [41,42]. OA is characterized by complex pathological mechanisms [15] and a broad range of clinical symptoms [15,43]. BMPs and GDF-5 play key roles in chondrocyte synthesis [8,9]. Compromised chondrogenesis promotes OA [10] and various components of the joint and/or synovial ECM, such as proteases and collagens, significantly contribute to the anabolism or catabolism of cartilage tissue and OA progression [10].

In early-stage OA, synovial inflammation results in synovitis [11], a condition characterized by numerous symptoms, including swelling and structural changes in the joints [2,11,12]. Synovitis-induced pathological alteration causes the release of proinflammatory cytokines such as IL-1β and TNF-α [13]. These inflammatory mediators are secreted by various types of cells, especially SFs in the joints [14,15]. Synovial fibrosis elicits the activation of many inflammatory cells, proinflammatory cytokines, and associated factors that facilitate OA development [44]. The balance of anabolic (the chondrogenesis inducers GDG-5, SOX5, TGF-β1, BMPR2, and COL4A1) [9] and catabolic (inflammatory mediators NFκB, TNF-α, and IL-6) [45] processes in the cartilage are essential for the maintenance of articular and cartilage health. Compromised synovial homeostasis and anabolic–catabolic imbalance in the joints result in OA progression [46]. This synovial–articular homeostasis is regulated and maintained by the crosstalk among several factors. According to one study, the catabolic process of cartilage degradation and OA progression is regulated by proinflammatory cytokines such as IL-6, TNF-α, and INF-γ [47]. NFκB, a key central inflammatory transcription factor, has also been implicated in the pathogenesis of OA [48]. Through a positive feedback loop, NFκB activates and is activated by inflammatory cytokines, such as a Disintegrin And Metalloproteinase with ThromboSpondin motif and MMPs, that drive the catabolic process in joints; such cytokines are key players in the destruction of cartilage [48]. In the present study, SC75741, a novel NFκB inhibitor, demonstrated anti-OA therapeutic potential; it effectively inhibited NFκB and strongly suppressed the expression of cytokines, chemokines, and proapoptotic factors [32] that facilitate the initiation and progression of OA. Some evidence indicates that miR’s are involved in synovitis and hyperplasia in the synovial fluid [27,28]. In a recent study, where articular cartilage explants were stimulated by IL-1β, differential expression of miR’s was observed in the knee synovial fluid of patients with OA [29]. Other studies have demonstrated the altered miR’s levels in the plasma of patients with OA [13,28]. The reduced expression of certain miR’s is indicative of their antiapoptotic activity and their enhancement of chondrocyte proliferation through NFκB expression regulation [29,30]. We demonstrated that IL-1β stimulated inflammation in the arthritic synovial membrane and activated α-SMA [46] rich synovial fibroblast-like chondrocytes. Moreover, coculturing AFs with human chondrocytes (C28/I2 and CA402) significantly enhanced the chondrocyte expression of the inflammatory markers IL-6, NFκB, TNF-α, and Cox2, all of which have been implicated in the induction of OA [49]. Our analysis of public databases and qRT-PCR analysis revealed that miR-21 is aberrantly expressed in dysfunctional AFs and ACs and that it targets genes involved in the maintenance of homeostasis of articular cartilage, including SOX5, GDF-5, TGFBR2, COL4A1, and SMAD7 [50,51]. This study also provides preclinical evidence to support the premise that the inhibition of miR-21 expression in IL-1β-stimulated AF-like SW982 cells results in the reduced expression of α-SMA. This indicates the suppression of a fibroblast-like phenotype, evidence for which was indicated by the suppressed migration of cells treated with the miR-21 inhibitor and the observation of the opposite effect in cells exposed to the miR-21 mimic.

As mentioned, bone formation and resorption form a continuum in articular cartilage homeostasis [52], maintaining the osteoblast–osteoclast balance and modulating the bone remodelling process [53]. Consistent with these premises, we demonstrated that ectopic miR-21 expression promoted osteoclast differentiation by increasing the population of multinucleated TRAP+ cells, as well as MMP9, OPG, and RANKL expression [54]. Furthermore, according to findings, concerning the role of miR-21 in the regulation of NFκB expression and activity [38,39], treatment with SC75741 substantially suppressed the expression of miR-21, α-SMA, and NFκB in OA-promoting both AFs and ACs cells, with concomitant downregulation of the key markers of hypertrophic chondrocytes and osteoclast differentiation.

In the in vivo rat model of chemically induced OA, SC75741 treatment was found to be effective. Specifically, compared with the corresponding measures in the rats with untreated OA, the severity of cartilage destruction in the SC75741-treated group, and the OARSI scores were also lower [55]. Furthermore, the expression of the apoptotic marker caspase-3 was downregulated in the treatment group. Consistent with reports that healthy cartilage exhibits lower IL-6 expression compared with cartilage in OA-affected joints [56], the present SC75741 treatment downregulated the expression of the proinflammatory cytokines IL-6, TNF-α, NFκB, and IL-1 β; catabolic enzyme MMP9; and serum miR-21 levels, protecting articular cartilage from further deterioration and preventing OA progression. SC75741 significantly suppressed the formation of mature multinuclear chondroclasts expressing the TRAP+ phenotype, indicating, at least in part, the reduction of RANKL-mediated chondroclast differentiation and activation [57] in rats with OA.

## 4. Materials and Methods

### 4.1. Microarray Datasets Search

The publicly available GEO databases (https://www.ncbi.nlm.nih.gov/geo/) accessed on 30 August 2021 [58] were searched using the keywords “miRNAs”, “fibroblast”, “dysfunction fibroblast”, and “osteoarthritis” to identify the key miR’s associated with dysfunctional fibroblasts and activated fibroblasts, and the GSE37627 datasets were used for further analysis. Differentially expressed miRNAs were obtained using the online tool GEO2R [58] accessed on 30 August 2021 in the GEO database by applying the fold change (|logFC| > 2) and *p*-value (<0.05) criteria.

### 4.2. Cell Culture and In Vitro Osteoclastogenesis

Human C20A4 and C28/I2 chondrocytes were maintained in a 1:1 mixture of Dulbecco’s Modified Eagle’s Medium (DMEM, GIBCO, Life Technologies Corp., Carlsbad, CA, USA) and F12 medium (GIBCO, Life Technologies Corp., Carlsbad, CA, USA) supplemented with 10% (v/v) fetal bovine serum (FBS, GIBCO, Life Technologies Corp., Carlsbad, CA, USA). Murine RAW264.7 and human synovial sarcoma SW982 cells were obtained from the American Type Culture Collection (ATCC, Manassas, VA, USA) and were cultured as per the recommendations of the ATCC. The culture medium was changed every 3 days until confluence; generally, cells were sub passage within 4–5 days.

The method suggested by Lampiasi, Nadia et al., 2021 with little modification [35] was applied in our study for osteoclasts (OC) differentiation from murine RAW 264.7 cells. In the brief to induce OC differentiation, the murine cells were suspended in an alpha-minimal essential medium (α-MEM Gibco, Grand Island) with 10% heat-inactivated fetal bovine serum (FBS, Sigma-Aldrich, USA), 100-U/mL penicillin, and 100-µg/mL streptomycin; after 24 h of the cultivation period, the previous α-MEM media was replaced with serum-free-α-MEM medium supplemented with 50-ng/mL RANKL-Recombinant-Protein (ThermoFisher, RP-8601) and human M-CSF (20 ng/mL; ThermoFisher, RP-8615). After 48 h, 20 µg/mL of osteopontin (OPG, Sigma-Aldrich, St. Louis, MO, USA) were added to various groups of cells in the presence of M-CSF and RANKL, and the cells were incubated for another 3 days.

### 4.3. Reagents

SC75741 N-(6-benzoyl-1H-benzo[d]imidazol-2-yl) -2-(1-(thieno[3,2-d] pyrimidine-4-yl) piperidine-4-yl) thiazole-4-carboxamide (MW 565), an NFκB inhibitor [54], which causes impaired DNA binding of the NFκB subunit p65, was obtained from Selleck Chemic (Houston, TX, USA) and dissolved in DMSO at a 50 mM stock concentration.

### 4.4. Cell Counting Kit-8 Cell Viability Assay

Cell proliferation was measured with the cell counting kit-8 (CCK-8) following the manufacturer’s instructions. After 48 h of transfection, human C20A4 and C28/I2 chondrocytes cells were seeded into a 96-well plate as per the manufacturer’s protocol. Then, 100 µL of 10% CCK-8 solution (Dojindo Laboratories, Kumamoto, Japan) was added into each well at different periods after treatments and incubated for 2 h at 37 °C. The absorbance at OD450 nm was measured by a microplate reader.

### 4.5. Total RNA Extraction and miR-21 Quantification by qRT-PCR and Transfections

The total RNA from cells and tissue samples after the respective treatment were isolated and purified using TRIzol-based protocol (Life Technologies) according to the manufacturer’s instructions. Five hundred nanograms of total RNA was reverse transcribed using QIAGEN One-step RT-PCR Kit (QIAGEN, Taipei, Taiwan), and the PCR reaction was performed using a Rotor-Gene SYBR Green PCR Kit (400, QIAGEN, Taiwan). The primers sequence for all the genes that are used in this study were purchased from QIAGEN (QIAGEN, Taipei, Taiwan), and detailed primer sequences are enumerated in Appendix A. For miR-21 quantification, total RNA was extracted from the cells, tissue, and blood samples using Trizol reagents (Invitrogen, Life Technologies) and by using PrimeScript RT Reagents Kit (Takara), and the first-strand cDNA was synthesized as per the manufacturer’s instructions. Relative miR-21 level in cells quantified by qRT-PCR used U6 miRNA as control. The qRT-PCR was carried out using a SYBR Premix Ex Taq Kit (Takara) on a QIAGEN rotor real-time PCR machine (QIAGEN, Valencia, CA, USA). The primers used for miR-21 amplification were (forward) 5′ACGTTGTGTAGCTTATCAGACTG3′ and (reverse) 5′AATGGTTGTTCTCCACACTCTC3′, and primers for U6 were (forward) 5′ATTGGAACGATACAGAGAAGATT3′ and (reverse) 5′GGAACGCTTCACGAATTTG3′. The overexpression and downregulation of miR-21 were achieved using mimic and inhibitor, respectively. MiR-21 mimic (HMI0372, Sigma, St. Louis, MO, USA) and inhibitor (HSTUD0371, Sigma, St. Louis, MO, USA) were transfected into the cells using the lipofectamine 2000 reagents (Invitrogen, USA) according to the vendor’s instructions.

### 4.6. Transwell Co-Culture and Migration Assays

For the co-culture system, the Transwell co-culture assay was performed; we first simulated the fibroblast-like SW982 synovial cell line with 10 ng/mL of pro-inflammatory cytokines, IL-1β for 24h. AF cells were co-cultured with chondrocytes cell lines (C20A4 and C28/I2). The AF cells were seeded at the upper chamber of the insert in a 6-well plate Transwell apparatus (0.4 µM pore size, Corning, Lowell, MA, USA), and the chondrocytes cell line was seeded in the lower chamber. After 48 h of co-culture induction, the chondrocytes were harvested by trypsinization. The migration assay was performed as per the protocol suggested by the manufacturer, and the transwell migration assay (ThermoFisher, Taipei, Taiwan) was applied to assess the migratory abilities of human C20A4 and C28/I2 chondrocytes cells. Each group of treated cells (1 × 10^5^ cells/mL) was diluted in (DMEM)–F12 medium without serum and seeded into the upper chamber. The lower chamber was filled with (DMEM)–F12 medium containing 10% FBS. After incubation at 37 °C for 24 h, the cells in the upper chambers were removed. Chondrocytes cells were fixed and stained with crystal violet. Then, randomly selected fields were imaged, and migrated cells were counted.

### 4.7. Western Blot Assay

Total proteins from the cells and tissue samples were extracted after treatment from different experiments were separated using a standard SDS-PAGE using the Mini-Protean III system (Bio-Rad, Taiwan) and western blotting using Trans-Blot Turbo Transfer System (Bio-Rad, Taiwan) was carried out according to the previously established method [59]. Membranes were incubated overnight at 4 °C in respective primary antibodies, and the secondary antibodies were purchased from Santa Cruz Biotechnology (Santa Cruz, CA, USA) and an ECL detection kit was used for the detection of the protein of interest. Images were captured and analyzed using UVP BioDoc-It system (Upland, CA, USA). Details of the primary antibody with the dilution and catalogue numbers are listed in Appendix A.

### 4.8. Immunofluorescence

For Immunofluorescence staining and analysis, the cells were plated in 6-well chamber slides (Nunc, Thermo Fisher Scientific, Taipei, Taiwan) at 4 °C overnight, fixed in 2% paraformaldehyde at room temperature for 10 min, then permeabilized with 0.1% Triton X-100 in 0.01 M phosphate-buffered saline (PBS), pH 7.4 containing 0.2% bovine serum albumin (BSA). Thereafter, they were air-dried and rehydrated in PBS, followed by incubating the cells with antibodies. For negative controls, we omitted the primary antibody. After PBS washing twice for 10 min each, anti-rabbit IgG fluorescein isothiocyanate-conjugated secondary antibody (Jackson Immunoresearch Lab. Inc., West Grove, PA, USA) diluted 1:500 in PBS was added, and the cells were incubated at room temperature for 1 h. This was followed by PBS washing and cell mounting using the Vectashield mounting medium and 4′, 6′-diamidino-2-phenylindole (DAPI) to counterstain DNA for nucleus visualization. Cells were observed under a Zeiss Axiophot (Carl Zeiss, Strasse, Oberkochen, Germany) fluorescence microscope.

### 4.9. Tartrate-Resistant Acid Phosphatase (TRAP) Staining

TRAP staining is considered to be a histochemical marker of osteoclasts. After being induced to differentiate into osteoclasts, the cells were fixed with 4% formaldehyde for 20 min at room temperature and then stained for TRAP (Sigma, St. Louis, MO, USA) according to the manufacturer’s instructions. As per the previously described procedure, TRAP staining was performed [60]. TRAP-positive multinuclear cells with 3 or more nuclei were counted as osteoclasts. TRAP-positive cells in these fields were counted by two blinded observers.

### 4.10. In Vivo OA Rat Model

Specific-Pathogen-Free Female Wistar rats (12 weeks old, 300–330 g) were purchased from BioLASCO Taiwan Co. Ltd. and used according to protocols approved by the Laboratory Animal Committee of the Taipei Medical University (protocol LAC-2020-0146). The OA model was prepared as per the previously described protocol by Murat et al. 2007 [61]. The 4% papain solution (200 µL) and 100 µL of 0.03 mol L^−1^ L-cysteine were prepared in distilled water. The reaction was mixed and incubated for 30 min. The mixture (25 µL) was injected into the right knee joint of the rat. Intra-articular injections were repeated on days 1, 4, and 7 to induce the OA. After the successful generation of OA (day 40), the rats were randomly divided into two groups consisting of 10 animals per group, where the disease control group received only vehicle (Saline) and the treatment group (SC75741). SC75741 (15 mg/kg) was administered orally thrice a week. The animal from each group was sacrificed on the last day (after 12 weeks) of the study, and knee samples and pooled blood samples from all groups of rats were analyzed for miR-21 plasma levels expression to evaluate disease progression.

### 4.11. Histopathology, Immunohistochemical Staining and Scoring

Wistar rats joints were collected after the treatment and fixed in 10% formalin solution with phosphate buffer saline (PBS) for 24 h. The tissues were dehydrated in the automatic dehydrator; conventional histopathological slides were collected with 4 µM sections. After dewaxing and dehydration of section, H&E staining or Safranin O/fast green staining was performed following the previously established protocol. After the staining, the pathological changes were observed using an optical microscope. Quick score (Q-score) was derived from the product of percentage (P) of tumor cells with characteristic IHC staining (0–100%) and the intensity (I) of IHC staining (1–3) (Q = P × I; maximum = 300) was used for the identification of IHC staining.

### 4.12. Osteoarthritis Research Society International (OARSI) Score

Safranin O Staining-based detection of the articular cartilage proteoglycan content. A well-accepted Osteoarthritis Research Society International (OARSI) scoring system was used in this study [62], and cartilage degradation was quantified by using the OARSI score. The total score was 24. The higher the score, the more serious the destruction of articular cartilage. The knee samples were harvested, and the tibiofemoral joints were removed. Then, the femoral condyles were fixed in 10% neutral-buffered formalin (containing 4% formaldehyde) for 24 h. After being washed with water for 2 h, the fixed femoral condyles were decalcified in 10% EDTA for 21 d. Then, graded ethanol dehydration, dimethyl benzene vitrification, paraffin embedding, and tissue sectioning (5 μm) were performed.

### 4.13. Statistical Analysis

Data are presented as mean ± SEM (standard error of the mean) of experiments performed at least 3 times in triplicates. Statistical analyses were performed using GraphPad Prism 7.0 (San Diego, CA, USA). Differences in Western blot band intensities between control and treatment groups are expressed as ±Δ%. The significance of single-point values was analyzed using chi-square (χ^2^) analysis, with the expected difference between control and treated groups set at Δ = ±20%. Comparison between two groups was performed using unpaired students’ *t*-test, while the comparison of groups ≥ 3 was carried out with the one-way analysis of variance (ANOVA) with Tukey’s posthoc multiple comparisons test. *p*-value < 0.05 was considered statistically significant.

## 5. Conclusions

OA is characterized by synovial inflammation—specifically, significantly elevated inflammation in the NFκB circuit (involving NFκB, IL-6, and TNF-α) with miR-21. Synovial inflammation promotes the generation of activated chondrocytes (ACs); ACs further promote inflammation through increased miR-21 signaling; the reduced expression of the chondrogenesis inducers SOX5, TGF-β1, and GDF-5 and the reduced expression of the ECM markers COL4A1 and ACAN. SC75741-induced inhibition of the NFκB circuit suppresses miR-21 signaling in activated chondrocytes and conversely upregulates its targets SOX5, TGF-β1, GDF-5, and COL4A1, preventing further articular chondrocyte destruction and mitigating OA overall (Figure 7). Our study has some limitations. As the human healthy synoviocytes, cell line or primary synoviocytes and patients’ samples has not been included in this study, which could be the best choice for this kind of study.

## Figures and Tables

**Figure 1 ijms-22-11082-f001:**
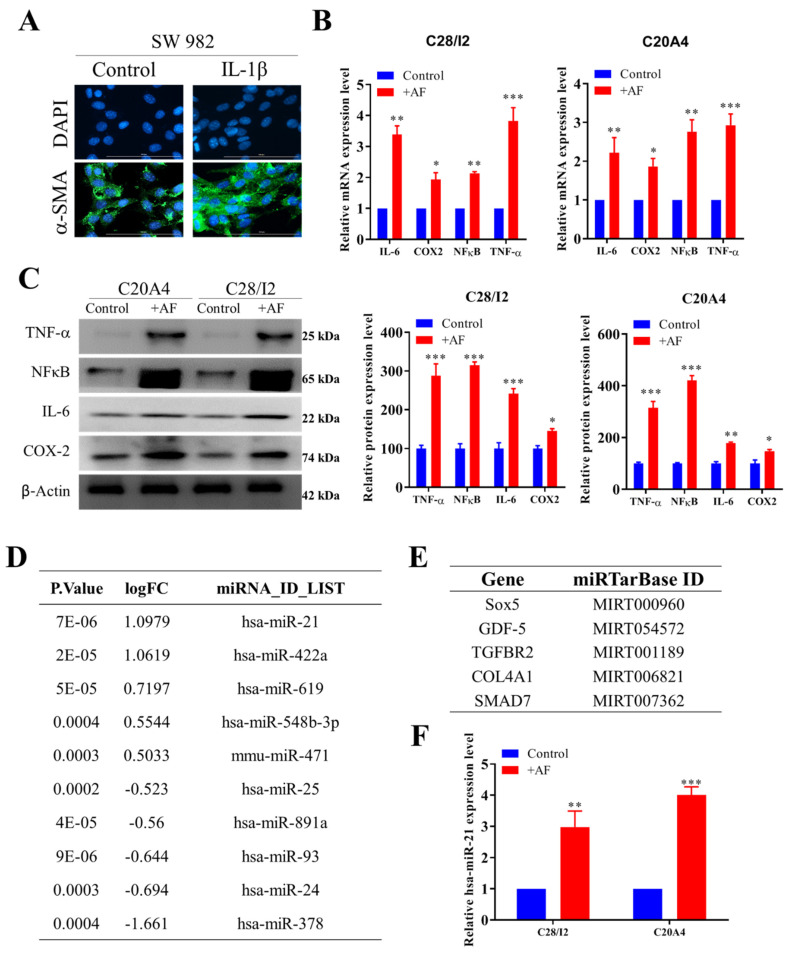
IL-1β signaling promotes the generation of activated fibroblasts. (**A**) Immunofluorescence analysis indicated that IL-1β treatment transformed SW982 cells into the AF phenotype. Green fluorescence indicates α-SMA labelling. (**B**) Human chondrocyte cell lines, C28/I2 and C20A4 exhibited increased levels of inflammatory markers after coculturing with AFs. (**C**) Comparative profiling between naïve chondrocytes and AF-cultured chondrocytes. Increased expression of TNF-α, NFκB, IL-6, and Cox2 was found in both C28/I2 and C20A4 cells after coculturing with AFs. (**D**) A bioinformatics search of the GSE37627 database identified miR-21 as one of the miRs with the greatest magnitude of increase in AFs compared with normal fibroblasts. (**E**) Experimentally validated targets of miR-21 in association with chondrogenesis. (**F**) Increased miR-21 levels in AF-educated C28/I2 and C20A4 chondrocytes. Scale bar = 50 μm. * *p* < 0.05, ** *p* < 0.01 and *** *p* < 0.001.

**Figure 2 ijms-22-11082-f002:**
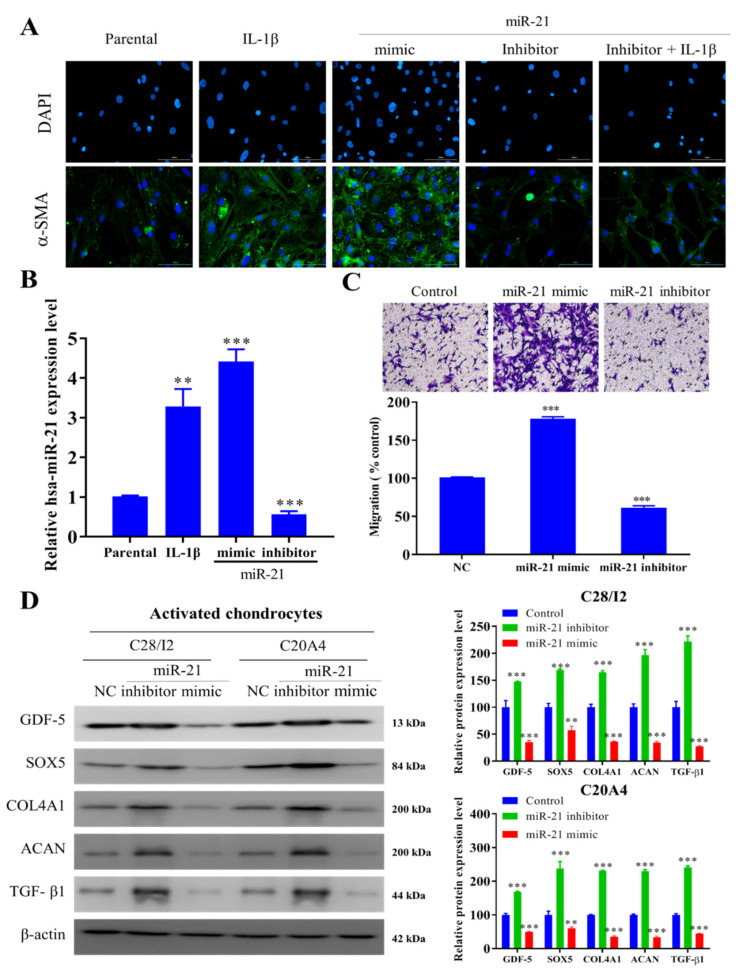
In-vitro inhibition of miR-21 checks the synovial fibroblast transformation. (**A**) Representative immunofluorescence image demonstrating the effects of miR-21 inhibition and overexpression on α-SMA expression in the presence or absence of IL-1β in SW982 cells. (**B**) qRT-PCR analysis of miR-21 expression after inhibition (Inhibitor), overexpression (Mimic), control (NC), and stimulation by IL-1β. (**C**) Representative image of cells after 24-h treatment demonstrating the effect of miR-21 on cellular motility. (**D**) Western blot and bar chart demonstrated protein expression of chondrogenic markers after miR-21 inhibition/overexpression to evaluate the effect of coculturing SW982 cells (AFs) with chondrocytes (C28/I2 and C20A4). Scale bar = 50 μm. ** *p* < 0.01 and *** *p* < 0.001.

**Figure 3 ijms-22-11082-f003:**
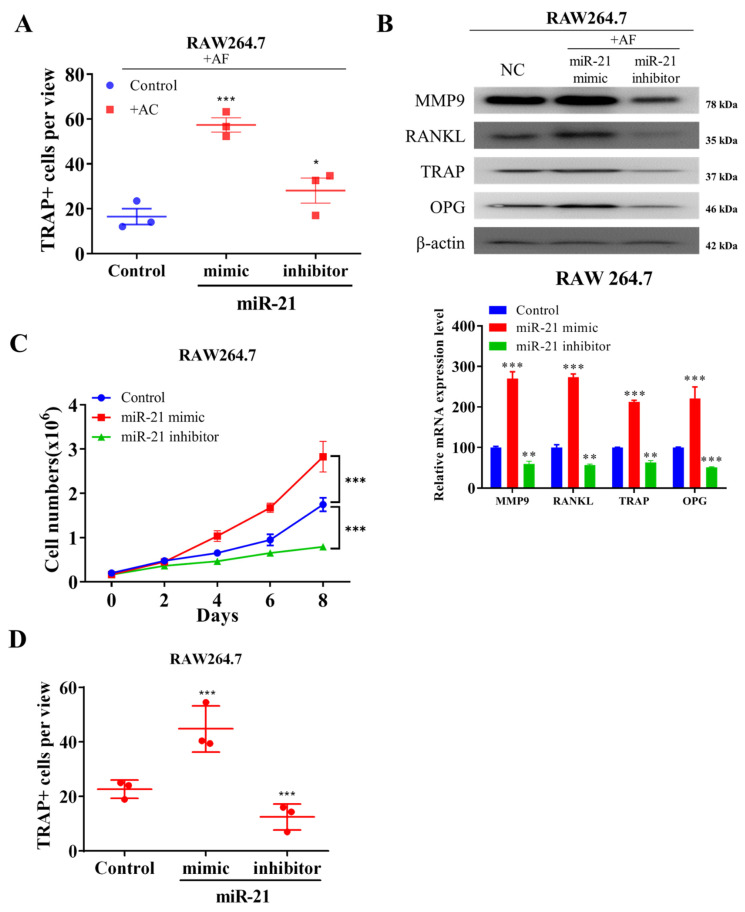
Overexpression of miR-21 in-vitro promotes the differentiation of osteoclasts. (**A**) Quantification of multinucleated TRAP+ RAW264.7 cells cocultured with miR-21-overexpressing and inhibited SW982 (AF) cells. (**B**) Western and bar plot demonstrated the expression of the level of osteoclast-specific genes in SW982 (AF) co-cultured RAW264.7 cells. (**C**) Effect of miR-21 overexpression and inhibition on the cell proliferative effect and the osteoclast differentiation of SW982 (AF) co-cultured RAW264.7 cells. (**D**) The osteoclast differentiation condition in presence of RANKL/M-CSF on miR-21-inhibited RAW264.7 cells. * *p* < 0.05, ** *p* < 0.01 and *** *p* < 0.001.

**Figure 4 ijms-22-11082-f004:**
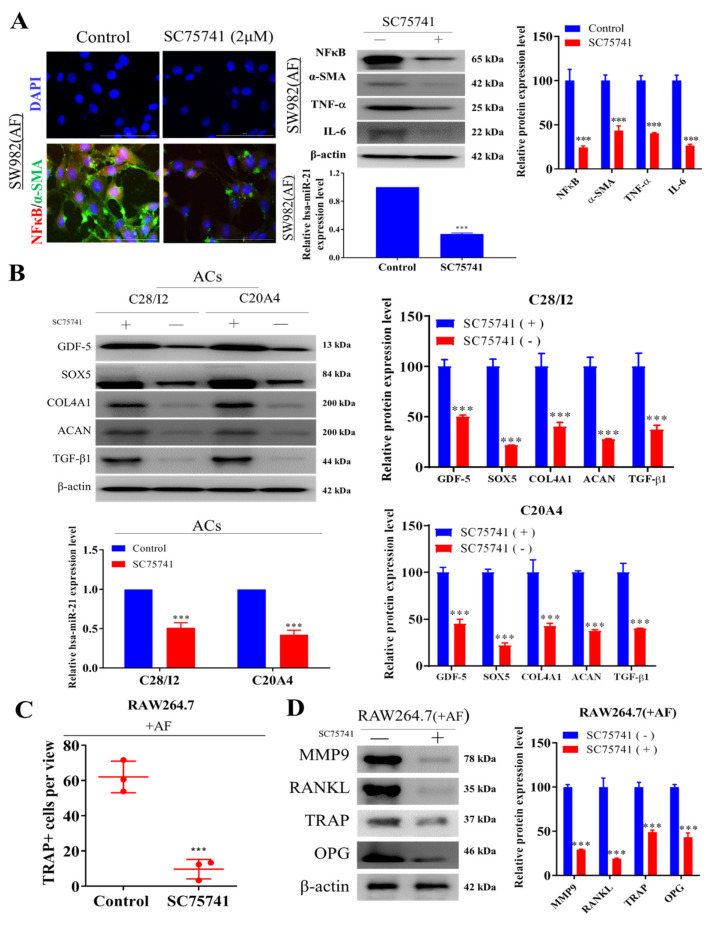
Effect of SC75741 a potent NFκB inhibitor on the transformation of activated fibroblasts, chondrocytes, and osteoclasts. (**A**) Representative immunofluorescence image demonstrating the effect of SC75741 on α-SMA and NFκB expression SW982 (AF) cells; qRT-PCR analysis of miR-21 expression and Western and bar blot analysis of expression of NFκB, α-SMA, TNFα, and IL-6 (at the protein level and normalized by the internal control β-actin) after SC75741 treatment. (**B**) Western and bar blot analysis of the expression of markers of human chondrocytes and qRT-PCR analysis of miR-21 expression in AC cells (C28/I2 and C20A4 chondrocytes). (**C**) TRAP staining of SC75741-treated AF cocultured with RAW264.7 cells. Quantification of multinucleated TRAP+ cells. (**D**) Protein analysis of the expression of osteoclast-specific genes in SC75741-treated SW982 cells (AF). Scale bar = 50 μm. *** *p* < 0.001.

**Figure 5 ijms-22-11082-f005:**
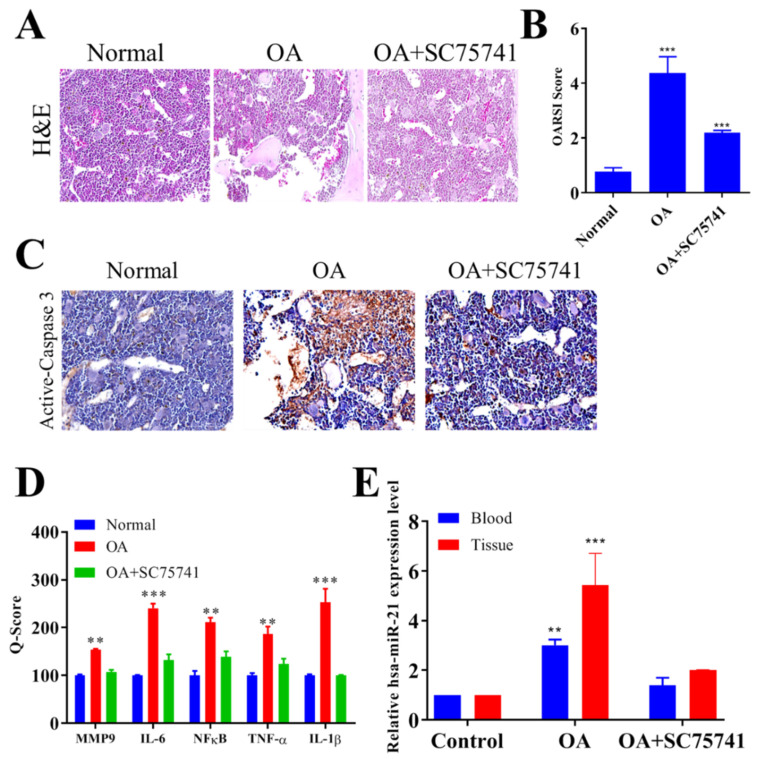
Histopathology of osteoarthritis of cartilage and osteoclastic activity after SC75741 treatment in the rat OA model. (**A**) H&E staining in each group at 6 weeks (magnification 20×). (**B**) OARSI score of each treatment group at 6 weeks. (**C**) IHC staining of cleaved caspase-3 expression in each group (magnification 40×). (**D**) SC75741 treatment reduced the expression of IL-6, TNF-α, and NFκB in the SC75741-treated rats with OA. (**E**) Quantification of miR-21 expression through qRT-PCR analysis. ** *p* < 0.01, and *** *p* < 0.001.

**Figure 6 ijms-22-11082-f006:**
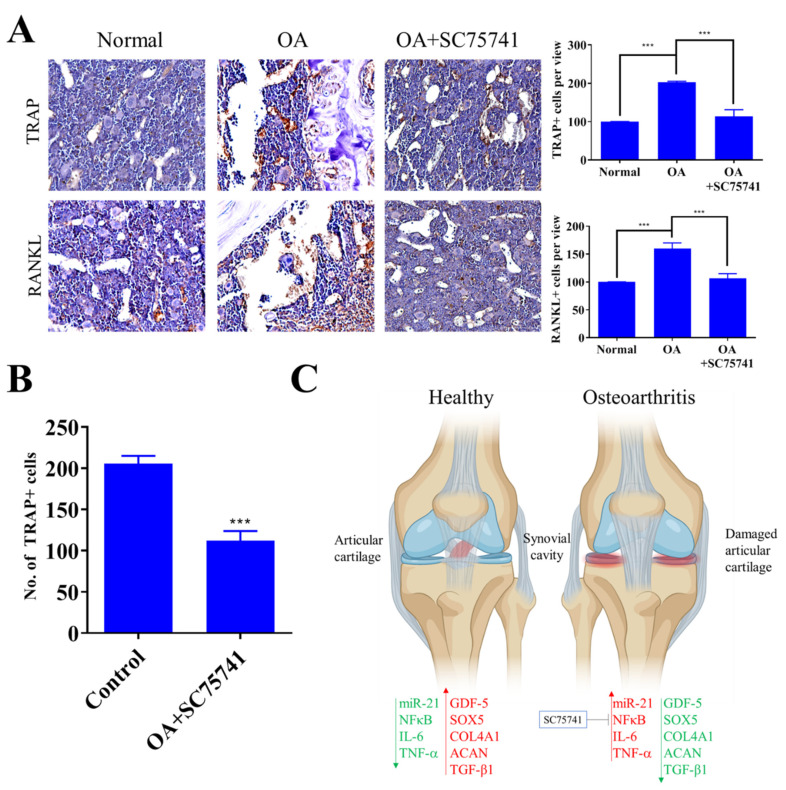
The effect of SC75741 on histopathology analysis showed a reduction of osteoclast formation and activity after SC75741 treatment in the rat OA model. (**A**) IHC staining was analyzed to evaluate the level of osteoclast in bone and cartilage. (**B**) The monocytes isolated from the joints of OA mice and control mice were cultured with 10ng/mL M-CSF and 2 μM SC75741 or 50 ng/mL RANKL. After two weeks of culture, the TRAP+ multinucleated cells were counted (at 10× magnification). (**C**) Schematic representation demonstrated SC75741-induced inhibition of the NFκB circuit suppressed miR-21 signaling in the activated chondrocytes while conversely upregulating its targets SOX5, TGF-β1, GDF-5, and COL4A1, thus consequently impeding further articular chondrocyte destruction and facilitating the amelioration of OA. *** *p* < 0.001.

**Figure 7 ijms-22-11082-f007:**
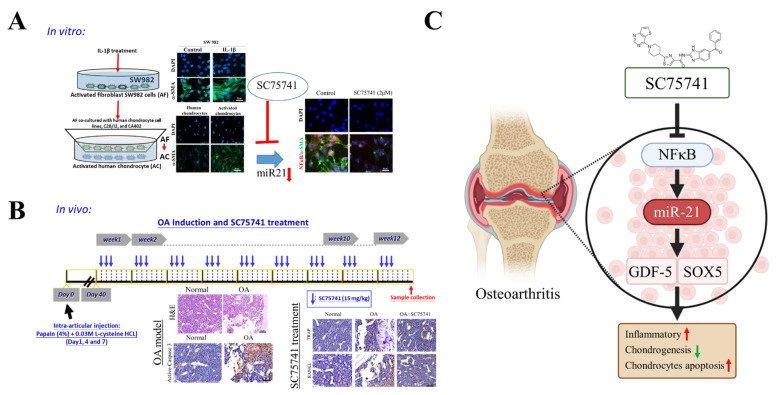
Overall study design. (**A**) Generation of AC cell model through co-culturing the activated fibroblast SW982 in presence of IL-1β and effect of SC75741 in reduction of miR-21 and NFκB. (**B**) Schematic representation of the timeline of chemically induced OA and SC75741 treatment. (**C**) Representation of the change in expression of key markers in OA progression and effect of SC75741 treatment. Scale bar = 50 μm.

## Data Availability

The datasets used and analyzed in the current study are publicly accessible as indicated in the manuscript.

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
