# Peer review of "Novel NFκB Inhibitor SC75741 Mitigates Chondrocyte Degradation and Prevents Activated Fibroblast Transformation by Modulating miR-21/GDF-5/SOX5 Signaling"

_ijms, 2021, doi:10.3390/ijms222011082_

Round 1

Reviewer 1 Report

In this experimental work both in vitro and in vivo, the authors evaluated the feasibility of using NFκB inhibitor, SC75741, for the possible treatment of Osteoarthritis. The study shows how SC75741 reduce OA severity and inhibits its progression. Osteoarthritis is a common and disabling disease and be able to stop the disease progression would be an important outcome for the patients. The article is interesting, clear and opens up new perspectives for the study of the disease.

I have some minor concerns:

In the abstract in the last lines of the results (between line 39 and 41) there are too many spaces between some words and in line 41 lacks there is a space missing between the two words "Sc75741significantly".

It would be nice a small schematic description of the experimental design that would be useful to the reader, in order to facilitate the understanding also to the less expert researchers in this field.

Author Response

We thank the reviewer for carefully reading our manuscript and providing valuable comments. We accordingly response the questions raised by the Reviewer as follows:

Point-by-point responses to reviewer’s comments:

Dear Reviewer,

Co-authors and I very much appreciated the encouraging, critical, and constructive comments on this manuscript by the reviewer. The comments have been very thorough and useful in improving the manuscript. We strongly believe that the comments and suggestions have increased the scientific value of the revised manuscript by many folds. We have taken them fully into account in revision. We are submitting the corrected manuscript with the suggestion incorporated in the manuscript. The manuscript has been revised as per the comments given by the reviewer, and our responses to all the comments are as follows:

Reviewer #1:

Comments and Suggestions for Authors

Journal: IJMS

[IJMS] Manuscript ID: ijms-1397959 - Major Revisions

Title: Novel NFκB Inhibitor SC75741 Mitigates Chondrocyte Degra-dation and Prevents Activated Fibroblast Transformation by Modulating miR-21/GDF-5/SOX5 Signaling

Authors: Pei-Wei Weng, Vijesh Kumar Yadav, Narpati Wesa Pikatan, Iat-Hang Fong, I-Hsin Lin*, Chi-Tai Yeh and Wei-Hwa Lee *

Revision:

In this experimental work both in vitro and in vivo, the authors evaluated the feasibility of using NFκB inhibitor, SC75741, for the possible treatment of Osteoarthritis. The study shows how SC75741 reduce OA severity and inhibits its progression. Osteoarthritis is a common and disabling disease and be able to stop the disease progression would be an important outcome for the patients. The article is interesting, clear and opens up new perspectives for the study of the disease.

A: We would like to thank the Reviewer for the thorough reading of our manuscript as well as the valuable comments and appraisal. We feel that the comments and suggestions will further be helped in strengthening our manuscript.

Q1: In the abstract in the last lines of the results (between line 39 and 41) there are too many spaces between some words and in line 41 lacks there is a space missing between the two words "Sc75741significantly".

A1:  We thank the reviewer for highlighting this point, we have rechecked all the typo, formatting mistakes in amended the current manuscript as per the suggestions.

Q2: It would be nice a small schematic description of the experimental design that would be useful to the reader, in order to facilitate the understanding also to the less expert researchers in this field.

A2:  We appreciate the reviewer's valuable suggestions and thank you for bringing up this good point, and we understand the reviewer’s viewpoint here and agree with this. In this current manuscript we have incorporated the overall design as per the reviewer’s suggestions in Graphical abstract Figure 7.

Reviewer 2 Report

The manuscript entitled “Novel NFκB Inhibitor SC75741 Mitigates Chondrocyte Degradation and Prevents Activated Fibroblast Transformation by Modulating miR-21/GDF-5/SOX5 Signaling” is an interesting study. In general, the manuscript is not easy to read. The authors have performed many experiments but the conditions used are not always clear. Not all protocols used have been reported in the methods creating doubts in the reader. The results are very difficult to read and to follow.

My comments are as follow:

1) First of all the authors use the following terms “Arthritic chondrocytes” and “AC-educated healthy chondrocytes (HCs)” without giving an clear explanation. Here, the study is focused on OA. Thus, using “arthritic chondrocytes” is confusing. Since C28/I2 and C20A4 are healthy chondrocytes cell lines, these should be simple called healthy chondrocytes. “AC-educated healthy chondrocytes” should be chondrocytes stimulated by activated fibroblasts. Thus, they could be called AF-stimulated chondrocytes to avoid confusion in the reader. (see also comment number 11)

2) The methods in the abstract should be revised. The authors should briefly report the methods used and not the clinical importance of the study.

3) The introduction on OA should be improved. The authors did not mention the recent role of infrapatellar fat pad, which is inflamed and fibrotic in OA patients (as well as synovial membrane). It has been suggested as infrapatellar fat pad and synovial membrane might act as an anatomo-functional unit. Moreover, the authors described the involvement of synovial inflammation and its consequences. However, they did not take into consideration that also the other joint tissues (meniscus, infrapatellar fat pad and cartilage itself) secretes cytokines etc (doi: 10.1080/03008207.2018.1470167), which could contribute to cartilage damage and also to synovial inflammation.

4) In the introduction, the authors reported “NFκB, a key inflammatory transcription factor, is highly expressed in patients with OA.”. This sentence is too vague. It should be better explained.

5) The last part of the introduction (from “to the best of our knowledge …”) is very confusing for the reader. It should be better rewritten. The aim/s of the study should be clear.

6) References should be cited in order of appearance. The authors should check.

7) Paragraph 2.4. and 2.6 should be merged in order to avoid to report the same method twice.

8) Paragraph 2.8: the authors reported that they differentiated the cells into osteoblast. The protocol should be added.

9) Paragraph 2.9: did the authors use female or male rats? “thrice” should be corrected. The authors reported that they sacrificed the animals on the last day of the study that means after how many days?

10) The authors reported that they stimulated fibroblast SW982 with IL-1beta as OA model. Fist, the protocol should be reported in the methods. Second, it is unclear why the authors selected a human synovial sarcoma cell line to mimic synoviocytes. This cell line is derived from a synovial sarcoma in the synovial membrane but this cancer is not related to synovial tissues. A human healthy synoviocytes cell line or primary synoviocytes would be the best choice for this kind of study. The authors should add this point as a study limitation.

11) Paragraph 3.1, the authors reported that they call the stimulated sarcoma synoviocytes as “AC”. I suppose that they mean “AF”. However, the authors reported in the discussion “IL-1β–stimulated SF-like SW982 cells (so-called AF or AC cells)”. Moreover, the reader has to wait until the conclusions to read “generation of activated chondrocytes (AFs/ACs)”. This is very confusing. I mean the authors used two abbreviations to define different stimulated cell lines. The authors should check and use only one abbreviation consistently for activated synoviocytes and one for chondrocytes stimulated with IL-beta treated synoviocytes.

12) The authors check only alpha sma to demonstrate fibroblast activation. However, activated fibroblast in OA are inflamed. Thus, it would be appropriate to quantify the expression of the principal cytokines.

13) Figure 1A-2A, 2C: the quality of the cell images should be improved. It is impossible to see the cells in the pdf. The authors should check.

14) The authors did not report the protocol/s used for immunofluorescence in the methods. Antibodies used for immunofluorescence are missing.

15) In the results the authors reported that they cultured chondrocytes in the presence of AF. Again the protocol used is missing. Did the authors use transwells? The detailed protocol should be reported.

16) The authors reported that they searched public databases for candidate miRs associated with dysfunctional fibroblasts in the results. How? What databases did the authors use? A brief paragraph regarding bioinformatics research should be added to the methods.

17) Figure 2.It is not clear to me why the authors introduced some abbreviations in the figure caption that are not used: “inhibition (I), overexpression (M)” instead of defining abbreviations used such as “NC”.

18) Pag.7: “sig-nificantlt” should be checked.

19) Pag. 7: “Furthermore, when co-cultured with miR-21 mimic bearing C28/I2 and C20A4 ACs exhibited reduced level of chondrogenesis markers, namely GDF-5, SOX5, TGF-β1, COL4A1, and ACAN proteins.” Did the authors transfect the cells with miR-21 mimic? If yes, using “co-cultured with miR-21 mimic” is confusing.

20) Figure 2D: it is unclear if chondrocytes were co-cultured with AF (synoviocytes) previous to be transfected with miR-mimic. Western blot bands should be quantified with a software like imageJ and a graph should be added.

21) It is unclear why the authors tested collagen IV instead of using collagen II , collagen I and collagen X.

23) Paragraph 3.3: “In the progressive development of OA, the balance between bone formation and re-sorption is significantly skewed towards the latter.” At least a reference should be provided.

24) Pag. 8: “Our investigation of the effects of miR-21 overexpression in ACs on osteoblast differentiation showed that co-culture of RAW264.7 macrophage cells with the activated fibroblast-like SW982 chondrocytes pre-exposed to miR-21 mimic significantly increased the population of macrophage-lineage TRAP+ cells, compared to their control or miR-21 inhibitor-exposed counterparts (Fig. 3A), suggesting that ectopic expression of miR-21 promotes the osteoblast-mediated differentiation of osteoclast precursors to osteoclasts.” This part should be better explained. First, it is not clear to me why the authors used a murine cell line (RAW264.7) and not a human cell line. It is not clear if the authors differentiated or not RAW264.7 cells before the co-culture. It is also not clear what is the control: RAW264.7 cells alone or RAW264.7 cells in co-culture with AF.

25) The authors should clearly explain the different conditions used in figure 3A and figure 3D.

26) Paragraph 3.4: “Exposure of IL-1β-induced ACs to 2μM SC75741 for 48 h resulted in markedly reduced the co-expression of α-SMA and NFκB proteins in the ACs (Fig. 4A, left).” . Again the use of “AC”  is very confusing. 

27) Figure 4A: It is not clear. Did the authors use also DAPI or Hoechst?

28) All the western blots should be quantified reporting the relative graphs with statistical analysis.

29) Pag. 9 :” Furthermore, when treated with 2μM SC75741 over-night, the population of TRAP+ ACs was significantly reduced (5.3-fold, p<0.001) (Fig. 4C), and this was associated with downregulated expression of MMP9, RANKL, TRAP, and OPG proteins (Fig. 4D), suggesting markedly reduced capability to induce osteoblast differentiation.” The authors should explain the experiment. It is for example unclear that the authors cocultured RAW264.7 cells with SC75741-treated AFs.

30) Figure 5D: “Q-score” is unclear.

31) In the rat model, it is not clear why the authors quantified miR21 expression in plasma. It would be important to quantify the mirRNA in the tissues. In the methods, the authors did not report that they collected plasma from the rats.

32) Paragraph 3.6: the authors reported that they co-cultured SC75741-pre-treated activated SW982 chondrocytes with bone marrow-derived monocytes (BMMs) in the presence of M-CSF and RANKL for two weeks. This part should be explained. Did the authors use C20A4 or C28/I2 chondrocytes? Did the authors co-cultured the chondrocytes with activated (with IL 1 beta) SW982 synoviocytes in the presence of SC75741? And then, the authors co-cultured stimulated chondrocytes with BMMs? The incubation times are not clear. Moreover, the reader understands that the authors isolated monocytes by reading figure 6 caption. The protocol should be clearly explained in the methods.

33) It would be useful to add line numbers in order to facilitate the revision of the paper.

Author Response

Reviewer #2:

The manuscript entitled “Novel NFκB Inhibitor SC75741 Mitigates Chondrocyte Degradation and Prevents Activated Fibroblast Transformation by Modulating miR-21/GDF-5/SOX5 Signaling” is an interesting study. In general, the manuscript is not easy to read. The authors have performed many experiments, but the conditions used are not always clear. Not all protocols used have been reported in the methods creating doubts in the reader. The results are very difficult to read and to follow.

A: We would like to thank the Reviewer for the thorough reading of our manuscript as well as the valuable comments. We have followed the reviewer's comments thoroughly and amended our current manuscript as per the suggestion to answers the questions asked. We feel that they have further helped in strengthening our manuscript.

Q1: First of all the authors use the following terms “Arthritic chondrocytes” and “AC-educated healthy chondrocytes (HCs)” without giving an clear explanation. Here, the study is focused on OA. Thus, using “arthritic chondrocytes” is confusing. Since C28/I2 and C20A4 are healthy chondrocytes cell lines, these should be simple called healthy chondrocytes. “AC-educated healthy chondrocytes” should be chondrocytes stimulated by activated fibroblasts. Thus, they could be called AF-stimulated chondrocytes to avoid confusion in the reader. (see also comment number 11)

A1:  We thank the reviewer for highlighting this point, we are sorry for the confusion on this point, we have edited and added more information about the term AF and AC in this currently updated manuscript. In brief, we first simulated the arthritic synovial microenvironment by culturing fibroblast-like SW982 synovial cell line with the inflammatory cytokine, IL-1β. The resultant SW982 cells (referred to as AF) showed markedly increased expression of α-SMA, a marker for activated fibroblast, compared to their unstimulated control counterparts (Fig. 1A). Subsequently, human chondrocyte cell lines, C28/I2, and CA402 were co-cultured in the presence of AF, referred to as activated chondrocytes (AC). Please kindly refer to page 3, line 122-127.

Q2: The methods in the abstract should be revised. The authors should briefly report the methods used and not the clinical importance of the study.

A2:  We appreciate the reviewer's valuable suggestions; in this updated manuscript we have briefly included the method section in the abstract. Please, kindly refer to the edited abstract part. Kindly refer to page 1, lines 28-39.

Q3: The introduction on OA should be improved. The authors did not mention the recent role of infrapatellar fat pad, which is inflamed and fibrotic in OA patients (as well as synovial membrane). It has been suggested as infrapatellar fat pad and synovial membrane might act as an anatomo-functional unit. Moreover, the authors described the involvement of synovial inflammation and its consequences. However, they did not take into consideration that also the other joint tissues (meniscus, infrapatellar fat pad and cartilage itself) secretes cytokines etc (doi: 10.1080/03008207.2018.1470167), which could contribute to cartilage damage and also to synovial inflammation.

A3: We again thank the reviewer for valuable suggestions on our manuscript. In this currently updated manuscript, we have discussed the infrapatellar fat pad and its association with OA. kindly see our revised manuscript main text in the introduction section, page 2, lines 82-90.

Q4: In the introduction, the authors reported “NFκB, a key inflammatory transcription factor, is highly expressed in patients with OA.”. This sentence is too vague. It should be better explained.

A4: We apologize for any confusion and thank you for the reviewer's insightful comments given here. As per the suggestions we also edited this sentence to make it clearer. Kindly refer to the main text introduction section on page 2, lines 100-102.

Q5: The last part of the introduction (from “to the best of our knowledge …”) is very confusing for the reader. It should be better rewritten. The aim/s of the study should be clear.

A5: As acknowledged by the reviewer, we again thank the reviewers for the valuable suggestions. We have edited the last part of the introduction to make it more clear for readers. Kindly, refer to the introduction section on page 3, lines 106-109.

Q6: References should be cited in order of appearance. The authors should check.

A6: Thank you very much for your insightful comments. We have checked and reordered the reference section in the order of appearance.

Q7: Paragraph 2.4. and 2.6 should be merged in order to avoid to report the same method twice.

A7: Thank you very much for your valuable comments. We have edited these materials and method sections and merged them. Please kindly refer to the materials and method section 4.5. of our newly edited manuscript. 

Q8: Paragraph 2.8: the authors reported that they differentiated the cells into osteoblast. The protocol should be added.

A8: We thank the reviewers for raising this key point, for studying the differentiation of cells, we applied the method suggested by Lampiasi, Nadia et.al, 2021 [1]. We have cited the reference of this method applied in our manuscript, kindly refer to page 6, line 175-179.

Q9:  Paragraph 2.9: did the authors use female or male rats? “thrice” should be corrected. The authors reported that they sacrificed the animals on the last day of the study that means after how many days?

A9: We sincerely thank the reviewer for the time taken to review our work, and for the valuable suggestions given. In this revised manuscript, we have rechecked and proofread all the typos as suggested by the reviewer. For this study, we used female mice, and the animals were sacrificed after 12 weeks of study.

Q10: The authors reported that they stimulated fibroblast SW982 with IL-1beta as OA model. Fist, the protocol should be reported in the methods. Second, it is unclear why the authors selected a human synovial sarcoma cell line to mimic synoviocytes. This cell line is derived from a synovial sarcoma in the synovial membrane, but this cancer is not related to synovial tissues. A human healthy synoviocytes cell line or primary synoviocytes would be the best choice for this kind of study. The authors should add this point as a study limitation.

A10: We thank the reviewers for this key point, as per the reviewer's suggestions, we have incorporated the details of the stimulated fibroblast SW982 with IL-1beta as OA model cells. In brief, for induction, we cultured SW982 cells with 10 ng/ml of pro-inflammatory cytokines IL-1beta for 24h, then these cells were used for further experiment. Please refer to page 3, lines 121-122 in our manuscript result part.

We also incorporated the study limitation, in our current manuscript regarding the human healthy synoviocytes cell line or primary synoviocytes usage.

Q11: Paragraph 3.1, the authors reported that they call the stimulated sarcoma synoviocytes as “AC”. I suppose that they mean “AF”. However, the authors reported in the discussion “IL-1β–stimulated SF-like SW982 cells (so-called AF or AC cells)”. Moreover, the reader has to wait until the conclusions to read “generation of activated chondrocytes (AFs/ACs)”. This is very confusing. I mean the authors used two abbreviations to define different stimulated cell lines. The authors should check and use

A11: We agree with reviewers’ suggestions, in this revised manuscript we have rechecked and used only one abbreviation consistently for activated synoviocytes and one for chondrocytes stimulated with IL-beta treated synoviocytes.

Q12: The authors check only alpha sma to demonstrate fibroblast activation. However, activated fibroblast in OA are inflamed. Thus, it would be appropriate to quantify the expression of the principal cytokines.

A12: Thank you very much for your insightful comments. As per the reviewer's suggestions, we have quantified the principal cytokines (TNFα and IL6) and incorporated the western blot data into Figure 4A (right).

Q13: Figure 1A-2A, 2C: the quality of the cell images should be improved. It is impossible to see the cells in the pdf. The authors should check.

A13: We appreciate the reviewer’s comment and apologize for the unclear and poor qualities of the images. In this revised version of the manuscript, we have incorporated the higher quality images. Kindly refer to the new added Figures 1A-2A, and 2C.

Q14: The authors did not report the protocol/s used for immunofluorescence in the methods. Antibodies used for immunofluorescence are missing.

A14: The reviewer’s comments are appreciated, and we apologize for not providing the protocols used for immunofluorescence in the methods. For Immunofluorescence staining and analysis, the cells were plated in 6-well chamber slides (Nunc, Thermo Fisher Scientific, Taipei, Taiwan) at 4 °C overnight, fixed in 2% paraformaldehyde at room temperature for 10 min, then permeabilized with 0.1% Triton X-100 in 0.01 M phosphate-buffered saline (PBS), pH 7.4 containing 0.2% bovine serum albumin (BSA). Thereafter, they were air-dried and rehydrated in PBS. Followed by incubating the cells with antibodies. For negative controls, we omitted the primary antibody. After PBS washing twice for 10 min each, anti-rabbit IgG fluorescein isothiocyanate-conjugated secondary antibody (Jackson Immunoresearch Lab. Inc., West Grove, PA, USA) diluted 1:500 in PBS was added, and the cells incubated at room temperature for 1 h. This was followed by PBS washing and cell mounting using the Vectashield mounting medium and 4′, 6′-diamidino-2-phenylindole (DAPI) to counterstain DNA for nucleus visualization. Cells were observed under a Zeiss Axiophot (Carl Zeiss, Strasse, Oberkochen Germany) fluorescence microscope. We have now added the aforementioned protocols in the materials and method section. Kindly, refer to the materials and methods section part (Page 14, section 4.8.).

Q15: In the results the authors reported that they cultured chondrocytes in the presence of AF. Again the protocol used is missing. Did the authors use transwells? The detailed protocol should be reported.

A15: We sincerely thank the reviewer for raising important points and valuable suggestions. We have used a cell co-culture system by Transwell cell coculture assay. In brief, AF was seeded at the upper-well of the trans-well plate and chondrocytes cell line was seeded at the bottom-well of the trans-well plate. After 48 h of induction, the chondrocytes were harvested by trypsinization. We have incorporated references and methods used for this part in our manuscripts materials and method section part on page 14, section 4.6.

Q16: The authors reported that they searched public databases for candidate miRs associated with dysfunctional fibroblasts in the results. How? What databases did the authors use? A brief paragraph regarding bioinformatics research should be added to the methods.

A16: We agree with reviewer’s suggestions, in this revised manuscript we have briefly included the bioinformatics research methodology. Kindly refer to the material and method sections 4.1. of our edited manuscript. 

Q17: Figure 2. It is not clear to me why the authors introduced some abbreviations in the figure caption that are not used: “inhibition (I), overexpression (M)” instead of defining abbreviations used such as “NC”.

A18: We sincerely thank the reviewer for the valuable suggestions. In this revised manuscript, we have rechecked the abbreviations and amended the changes as the suggestions. Kindly refer to Figure 2 legend in the result sections.

Q18: Pag.7: “sig-nificantlt” should be checked.

A18: We sincerely thank the reviewer for the time taken to review our work, and for the valuable suggestions given. In this revised manuscript, we have rechecked and proofread all the typos as suggested by the reviewer.

Q19: Pag. 7: “Furthermore, when co-cultured with miR-21 mimic bearing C28/I2 and C20A4 ACs exhibited reduced level of chondrogenesis markers, namely GDF-5, SOX5, TGF-β1, COL4A1, and ACAN proteins.” Did the authors transfect the cells with miR-21 mimic? If yes, using “co-cultured with miR-21 mimic” is confusing.

A19: We highly appreciate the reviewers’ insightful and helpful comments on our manuscript and apologize for the confusion. Sorry for our over-sight, We have transfected these ACs with miR21 (mimic and inhibitor) and NT (control), and after the transfection, we examined the expression of GDF-5, SOX5, TGF-β1, COL4A1, and ACAN proteins. As miR21 has been reported[2] to controls the development of OA by targeting the aforementioned markers in chondrocytes, we can see with the overexpression of miR21 (mimic) these markers were downregulated, vice versa in the case of miR21 inhibition (inhibitor).

Q20: Figure 2D: it is unclear if chondrocytes were co-cultured with AF (synoviocytes) previous to be transfected with miR-mimic. Western blot bands should be quantified with a software like imageJ and a graph should be added.

A20: We thank the reviewer for this insightful suggestion and comments, yes, these chondrocytes were activated in presence of AF before transfection. As suggested by the reviewer, we have quantified the western blot data and incorporated the bar plot. Please kindly refer to our revised attached quantified western blot results and bar plot figure in the result section of our newly attached manuscript.  

Q21: It is unclear why the authors tested collagen IV instead of using collagen II , collagen I and collagen X.

A21: The reviewer's insightful comments are greatly appreciated. As per the bioinformatics study conducted by Peiheng He et. al 2016 [3] “Screening of gene signatures for rheumatoid arthritis and osteoarthritis based on bioinformatics analysis”, along with COL3A1, COL1A1 and COL11A1, COL4A1 is also a key target involved in the OA and as per novelty and less study reported its association with OA, we examined the expression of COL4A1, and discussed in our manuscript.

Q22: Paragraph 3.3: “In the progressive development of OA, the balance between bone formation and re-sorption is significantly skewed towards the latter.” At least a reference should be provided.

A22. We thank you for the suggestions, in this revised manuscript we have incorporated the cited reference for this sentence. Kindly refer to page 6, line 169. 

Ji, B.; Zhang, Z.; Guo, W.; Ma, H.; Xu, B.; Mu, W.; Amat, A.; Cao, L. Isoliquiritigenin blunts osteoarthritis by inhibition of bone resorption and angiogenesis in subchondral bone. Scientific Reports 2018, 8, 1721.

Q23: Pag. 8: “Our investigation of the effects of miR-21 overexpression in ACs on osteoblast differentiation showed that co-culture of RAW264.7 macrophage cells with the activated fibroblast-like SW982 chondrocytes pre-exposed to miR-21 mimic significantly increased the population of macrophage-lineage TRAP+ cells, compared to their control or miR-21 inhibitor-exposed counterparts (Fig. 3A), suggesting that ectopic expression of miR-21 promotes the osteoblast-mediated differentiation of osteoclast precursors to osteoclasts.” This part should be better explained. First, it is not clear to me why the authors used a murine cell line (RAW264.7) and not a human cell line. It is not clear if the authors differentiated or not RAW264.7 cells before the co-culture. It is also not clear what is the control: RAW264.7 cells alone or RAW264.7 cells in co-culture with AF.

A23:  The reviewer’s comments are appreciated, and we apologize for the confusion, the RAW 264.7 cells, which is a murine monocyte-macrophage cell line, provide a valuable and widely used tool for in vitro studies on osteoclastogenesis, as per the methods suggested by Lampiasi, Nadia et.al, 2021 [1], were applied in our study for osteoclasts differentiation from murine RAW 264.7 cells stimulated by RANKL. RAW264.7 cells coculture with miR-21 overexpressing and inhibited SW982(AF) cells. Please kindly refer to our revised manuscript to the result section, on page 6, and lines 171-175.   

Q24: The authors should clearly explain the different conditions used in figure 3A and figure 3D.

A24: We highly appreciate the reviewers’ comments on our manuscript and apologize for the confusion. For figure 3A the RAW264.7 cells were co-culture with miR21 inhibited or overexpressed +AF, and in figure 3D the RAW264.7 non-induced cells were used to show the osteoclast differentiation condition in presence of RANKL/M-CSF on miR-21 inhibited RAW264.7 cells.

Q25: Paragraph 3.4: “Exposure of IL-1β-induced ACs to 2μM SC75741 for 48 h resulted in markedly reduced the co-expression of α-SMA and NFκB proteins in the ACs (Fig. 4A, left).” . Again the use of “AC”  is very confusing.

A25: The reviewer’s comments are appreciated and we apologize for the confusion we have rechecked and amended the changes as per the suggestion. Please kindly refer to our revised manuscript to the result section, on page 8, and lines 204-206.   

Q26: Figure 4A: It is not clear. Did the authors use also DAPI or Hoechst?

A26: We highly appreciate the reviewers’ insightful and helpful comments on our manuscript and apologize for the confusion. For the immunofluorescence experiment, we have used DAPI dye.

Q27: All the western blots should be quantified reporting the relative graphs with statistical analysis.

A28: We thank the reviewer for this insightful suggestion. As suggested by the reviewer, we have quantified the western blot data and incorporated the bar plot. Please kindly refer to our revised attached quantified western blot results and bar plot figure in the result section of our newly attached manuscript.  

Q28: Pag. 9 :” Furthermore, when treated with 2μM SC75741 over-night, the population of TRAP+ ACs was significantly reduced (5.3-fold, p<0.001) (Fig. 4C), and this was associated with downregulated expression of MMP9, RANKL, TRAP, and OPG proteins (Fig. 4D), suggesting markedly reduced capability to induce osteoblast differentiation.” The authors should explain the experiment. It is for example unclear that the authors cocultured RAW264.7 cells with SC75741-treated AFs.

A28: We highly appreciate the reviewers’ comments on our manuscript and apologize for the confusion. For figure 4A the RAW264.7 cells were co-culture with 2uM of SC75741 treated and non-treated +AF cells, after overnight induction the cell were harvested to check the expression of MMP9, RANKL, TRAP, and OPG proteins.

Q29: Figure 5D: “Q-score” is unclear.

A29: Again, thanks to the reviewer for the valuable suggestions. The quick (Q)-score used for quantification of immunohistochemical staining using the formula Q = P × I, where P is the percentage of positively stained cells and I is the intensity of staining. The maximum Q-score was 300. We have incorporated these details in the main text, kindly refer to the materials method section, on page 15, and lines 470-473.  

Q30:  In the rat model, it is not clear why the authors quantified miR21 expression in plasma. It would be important to quantify the mirRNA in the tissues. In the methods, the authors did not report that they collected plasma from the rats.

A30: We highly appreciate the reviewers’ insightful and helpful comments on our manuscript, we also agree with the reviewer's comments and as per the suggestions we have assayed and incorporated the expression profile of miR21 from tissue samples and also described in the materials and methods section about the collection of plasma from animals. Please kindly refer to Figure 5E in the result section and the method of the material section on page 15, lines 460-463.

Q31: Paragraph 3.6: the authors reported that they co-cultured SC75741-pre-treated activated SW982 chondrocytes with bone marrow-derived monocytes (BMMs) in the presence of M-CSF and RANKL for two weeks. This part should be explained. Did the authors use C20A4 or C28/I2 chondrocytes? Did the authors co-cultured the chondrocytes with activated (with IL 1 beta) SW982 synoviocytes in the presence of SC75741? And then, the authors co-cultured stimulated chondrocytes with BMMs? The incubation times are not clear. Moreover, the reader understands that the authors isolated monocytes by reading figure 6 caption. The protocol should be clearly explained in the methods.

A31: We highly appreciate the reviewers’ comments on our manuscript and apologize for the confusion and our oversight. To assess the effect of SC75741 on RANKL-induced chondroclast differentiation, bone marrow-derived monocytes (BMMs) were cultured with SC75741 in the presence of M-CSF and RANKL for 2 weeks, as per the method suggested by Kwon JY et.al 2018 [4]. SC75741 significantly suppressed the effect on RANKL-mediated chondroclast differentiation and activation. We have corrected the sentence, kindly refer to the result section 2.6.

Q32: It would be useful to add line numbers in order to facilitate the revision of the paper.

A31: Thank you very much for your insightful comments. We have added the line number in this newly edited manuscript Please kindly refer to the new attached edited manuscript.

References:

  1. Lampiasi, N., et al., Osteoclasts Differentiation from Murine RAW 264.7 Cells Stimulated by RANKL: Timing and Behavior. 2021. 10(2): p. 117.
  2. Zhang, Y., et al., MicroRNA-21 controls the development of osteoarthritis by targeting GDF-5 in chondrocytes. Experimental & Molecular Medicine, 2014. 46(2): p. e79-e79.
  3. He, P., et al., Screening of gene signatures for rheumatoid arthritis and osteoarthritis based on bioinformatics analysis. Molecular medicine reports, 2016. 14(2): p. 1587-1593.
  4. Kwon, J.Y., et al. Kartogenin inhibits pain behavior, chondrocyte inflammation, and attenuates osteoarthritis progression in mice through induction of IL-10. Scientific reports, 2018. 8, 13832 DOI: 10.1038/s41598-018-32206-7.

Round 2

Reviewer 2 Report

The manuscript improved after the revision. However, I have still some comments for the authors.

1) I understand that the authors used the protocol reported by  Lampiasi, Nadia et.al, 2021 for the differentiation of cells. However, it should briefly report the protocol in the methods (specifying also the concentration, company of RANKL etc).  

2) In the previous report, I suggested to quantify the western blot images and report the relative graphs. In the revised version, the authors quantified the bands reporting numbers on the bands but did not add the graphs. The authors should delete numbers on the bands and add the graphs (like the one in figure 2B) reporting the quantification of the proteins with appropriate statistical analysis.  

Author Response

We thank the reviewer for carefully reading our manuscript and providing valuable comments. We accordingly response the questions raised by the Reviewer as follows:

Point-by-point responses to reviewer’s comments:

 :::::::::::::::::::::::::::::::::::::::::::::::::::::::::::::::::::::::::::::::::::::::::::::::::::::::::::::::::::::::::::::::::::::::

Dear Reviewer,

Co-authors and I very much appreciated the encouraging, critical, and constructive comments on this manuscript by the reviewer. The comments have been very thorough and useful in improving the manuscript. We strongly believe that the comments and suggestions have increased the scientific value of the revised manuscript by many folds. We have taken them fully into account in revision. We are submitting the corrected manuscript with the suggestion incorporated in the manuscript. The manuscript has been revised as per the comments given by the reviewer, and our responses to all the comments are as follows:

:::::::::::::::::::::::::::::::::::::::::::::::::::::::::::::::::::::::::::::::::::::::::::::::::::::::::::::::::::::::::::::::::::::::

Reviewer #2:

Q1: 1) I understand that the authors used the protocol reported by Lampiasi, Nadia et.al, 2021 for the differentiation of cells. However, it should briefly report the protocol in the methods (specifying also the concentration, company of RANKL etc). 

A1: The reviewer’s comments are appreciated, and we apologize for not providing detailed protocols, in this revised manuscript we have included the detailed protocol for the differentiation of cells. Kindly refer to the material and method section, page 16, lines 364-381.   

4.2. Cell culture and In vitro Osteoclastogenesis

Human C20A4 and C28/I2 chondrocytes were maintained in a 1:1 mixture of Dulbecco’s Modified Eagle’s Medium (DMEM) and F12 medium supplemented with 10% (v/v) fetal bovine serum (FBS). Murine RAW264.7 and human synovial sarcoma SW982 cells were obtained from the American Type Culture Collection (ATCC) and were cultured as per the recommendations of the ATCC. The culture medium was changed every 3 days until confluence; generally, cells were sub passage within 4–5 days.

The method suggested by Lampiasi, Nadia et.al, 2021 with little modification[38], were applied in our study for osteoclasts (OC) differentiation from murine RAW 264.7 cells. In the brief to induce OC differentiation, the murine cells were suspended in an alpha-minimal essential medium (α-MEM Gibco, Grand Island) with 10% heat-inactivated fetal bovine serum (FBS, Sigma-Aldrich, USA), 100-U/mL penicillin, and 100-µg/mL streptomycin, after 24 h of the cultivation period, the previous α-MEM media was replaced with serum-free-α-MEM medium supplemented with 50-ng/mL RANKL-Recombinant-Protein (ThermoFisher, RP-8601) and human M-CSF (20 ng/ml; ThermoFisher, RP-8615). After 48 h, 20µg/ml osteopontin (OPG, Sigma-Aldrich, St. Louis, MO, USA) were added to various groups of cells in the presence of M-CSF and RANKL, and the cells were incubated for another 3 days.

2) In the previous report, I suggested to quantify the western blot images and report the relative graphs. In the revised version, the authors quantified the bands reporting numbers on the bands but did not add the graphs. The authors should delete numbers on the bands and add the graphs (like the one in figure 2B) reporting the quantification of the proteins with appropriate statistical analysis. 

A2: We have followed the reviewer's comments thoroughly and amended our current manuscript as per the suggestions. As suggested by the reviewer, we have quantified the western blot data and incorporated the bar plot. Please kindly refer to our revised attached quantified western blot results and bar plot figures with statistical significance in the result section.

:::::::::::::::::::::::::::::::::::::::::::::::::::::::::::::::::::::::::::::::::::::::::::::::::::::::::::::::::::::::::::::::::::::::
